# The Origin of the Non-Constancy of the Bulk Resistance of Ion-Selective Electrode Membranes within the Nernstian Response Range

**DOI:** 10.3390/membranes11050344

**Published:** 2021-05-07

**Authors:** Valentina Keresten, Elena Solovyeva, Konstantin Mikhelson

**Affiliations:** Chemistry Institute, Saint Petersburg State University, 26 Universitetsky Prospect, 198504 Saint Petersburg, Russia; v_lukina@list.ru (V.K.); solovyeva.elena.v@gmail.com (E.S.)

**Keywords:** ion-selective membranes, ionophores, mixed aqueous solutions of electrolytes, resistance, water uptake, nanometer-sized water droplets, potentiometry, chronopotentiometry, electrochemical impedance, gravimetry

## Abstract

The dependence of the bulk resistance of membranes of ionophore-based ion-selective electrodes (ISEs) on the composition of mixed electrolyte solutions, within the range of the Nernstian potentiometric response, is studied by chronopotentiometric and impedance measurements. In parallel to the resistance, water uptake by the membranes is also studied gravimetrically. The similarity of the respective curves is registered and explained in terms of heterogeneity of the membranes due to the presence of dispersed aqueous phase (water droplets). It is concluded that the electrochemical equilibrium is established between aqueous solution and the continuous organic phase, while the resistance refers to the membrane as whole, and water droplets hamper the charge transfer across the membranes. In this way, it is explained why the membrane bulk resistance is not constant within the range of the Nernstian potentiometric response of ISEs.

## 1. Introduction

Ion-selective electrodes (ISEs) with solvent-polymeric membranes (most commonly made of plasticized polyvinylchloride, PVC) and containing ionophores and ion exchangers (ionic additives) are routinely used as potentiometric (zero-current) sensors in a large variety of applications [1,2,3]. Analysis with ISEs in non-zero current modes of measurements, e.g., voltammetry [4,5,6,7] and coulometry [8,9,10,11,12,13], offers new advantages. In particular, the voltammetric mode of measurements allows for quantification of several analyte ions with a sole electrode [5,6,7], while coulometric measurements provide improved sensitivity, which is especially important in clinical applications [9].

Although non-zero current measurements with ISEs are promising for analytical applications, zero-current (potentiometric) measurements with ISEs remain the most common mode of their practical use. In turn, practical use of ISEs is largely limited to measurements within the Nernstian response range.

According to the established views on how ISEs work, within the linear Nernstian response range, only the boundary potential at the membrane/sample interface varies. Significant variation of other contributions to the overall membrane potential takes place only outside the Nernstian response range. If the interface between the membrane and solution is at electrochemical equilibrium, the boundary potential obeys the equation (1):(1)$$\emptyset_{b}=\frac{\mu_{1}^{0,aq}-\mu_{1}^{0,mem}}{F}+\frac{RT}{z_{1}F}\ln\frac{a_{1}^{aq}}{a_{1}^{mem}}$$

Here, $\emptyset_{b}$ is the boundary potential, $\mu_{1}^{0,aq},\;\mu_{1}^{0,mem}$ are the standard chemical potential values, $a_{1}^{aq},\;a_{1}^{mem}$ are the activities of $I^{Z_{I}}$ analyte ion in the aqueous phase and in the membrane phase, and $z_{I}$ is the ion charge. An ISE only shows a Nernstian response to $a_{I}{}^{aq}$ activity of $I^{Z_{I}}$ analyte ion in solution if $a_{I}{}^{mem}$: the activity of $I^{Z_{I}}$ analyte ion in the membrane phase is constant. This is ensured by the addition of ion exchangers (ionic additives) to the ISE membranes. As a result, the ISE membranes interact with solutions only via ion exchange (co-extraction of the electrolyte from solution and transmembrane fluxes are negligible). A constant value of $a_{I}{}^{mem}$ suggests a constant composition of the membrane, and therefore, within the Nernstian response range, one should expect a constant resistance of the membrane bulk.

However, our recent chronopotentiometric and impedance experiments with ISEs selective to Ca^2+^ [14], NO_3_^−^ [15], K^+^ [16] and Cd^2+^ ions [17] revealed non-constancy of the bulk resistance of the ISE membranes within the Nernstian response range. All four kinds of ISEs showed an increase of the membrane bulk resistance along with the decrease of the concentration of the respective electrolytes (CaCl_2_, KNO_3_, KCl and CdCl_2_) below 0.01 M. These data appear in conflict with the reasoning above [18,19,20].

Measurements of water uptake by the membranes showed that water uptake also increases along with a decrease of the electrolyte concentration, and the respective curves—resistance vs. concentration and water uptake vs. concentration—are similar [17,21]. This similarity implies that the resistance is somehow determined by water uptake.

It was suggested that water uptake by ISE membranes is driven by the osmotic pressure of aqueous solution [22,23]. Osmotic pressure is a colligative property. Therefore, for an ISE selective to, e.g., ion *I*, water uptake and membrane bulk resistance must depend on the total concentration of the electrolytes in mixed solutions or on the ionic strength, rather than on the concentration of this ion. Here, for the first time, we report on a systematic study of the bulk resistance of ISE membranes in contact with mixed electrolyte solutions. In parallel to the resistance, we also measured water uptake. ISEs selective to K^+^, Ca^2+^, Cd^2+^, and NO_3_^−^ ions were chosen as a model system because (I) these ISEs are of high practical relevance and (II) the dependence of the resistance of these electrodes in the respective pure electrolyte solutions was revealed in our former studies [14,15,16,17,21].

## 2. Materials and Methods

Neutral ionophores potassium ionophore I: valinomycin, calcium ionophore I: diethyl N,N’-[(4R,5R)-4,5-dimethyl-1,8-dioxo-3,6-dioxaoctamethylene]bis(12-methylaminododecanoate)] (ETH 1001), cadmium ionophore I: *N*,*N*,*N*′,*N*′-tetrabutyl-3,6-dioxaoctanedi(thioamide) (ETH 1062), cation exchanger potassium tetrakis-*p*-Cl-phenylborate (KClTPB), plasticizers bis(2-butylpentyl)adipate (BBPA), bis(2-ethylhexyl)phthalate (DOP), and 2-nitrophenyloctylether (oNPOE), and high molecular weight poly(vinyl chloride) (PVC) were Selectophore-grade reagents from Merck (Darmstadt, Germany). Anion exchanger tetradecylammonium bromide (TDABr) was from Analiz-X (Minsk, Belarus). Volatile solvents extra pure cyclohexanone (CH) and HPLC grade tetrahydrofuran (THF), as well as acetonitrile, were from Vekton (St. Petersburg, Russia). Ethylendioxythiophene (EDOT) was from FluoroChem (Glossop, UK). Sodium polystyrensulfonate (NaPSS) was from Aldrich (Irvine, CA, USA). Inorganic salts were from Reaktiv (Moscow, Russia). All aqueous solutions were prepared with deionized water with resistivity 18.2 MOhm∙cm (Milli-Q Reference, Millipore, France). Glassy carbon rods with diameter of 3 mm, encapsulated in Teflon bodies with diameter of 7 mm, were from Volta (St. Petersburg, Russia). Diamond slurry (1 µm) was from Antec Scientific (Zoeterwoude, The Netherlands); alumina 0.3 µm suspension was from Buehler (Lake Bluff, IL, USA).

The membrane cocktails were prepared by dissolving appropriate amounts of PVC, plasticizer, ionophore, and ion exchanger in THF. The percentage of “dry” content in the cocktails was 18%. To obtain the membranes, the cocktails were stirred for 30 min using roller-mixer Selecta Movil Rod (Barcelona, Spain) and then cast on glass Petri dishes. The dishes were closed with filter paper to slow down the evaporation of THF. The complete evaporation of THF took 1 day, and after that, master membranes with a thickness of about 0.4 mm were obtained. Four membranes were prepared; for the membrane compositions, see Table 1.

Two types of electrodes were used in the study: classical ISEs with an internal solution and an internal Ag/AgCl electrode, and solid-contact ISEs with PEDOT-PSS in the transducer layer. Membranes selective to K^+^, Ca^2+^, Cd^2+^, and NO_3_^−^ ions were studied in the classical construct, while Cd^2+^-ISEs were also of the solid-contact type.

Classical ISEs were prepared by cutting disks with a diameter of 12 mm from the master membrane and gluing them to PVC bodies with an outer diameter of 12 mm and an inner diameter of 10 mm. A solution of PVC in CH was used as the glue. The internal reference electrode was chlorinated silver wire in a polypropylene body. The electrode constructs are shown in Appendix A.

Initially, K-, Ca-, and Cd-ISEs were filled with 0.01 M solutions of the respective chlorides, and NO_3_-ISE with 0.01 M KNO_3_, and conditioned in the same solutions for 2 weeks. In the case of NO_3_-ISEs, the internal solution and the soaking solution were replaced with a fresh solution every other day. This was done to replace Br^−^ in the membrane phase with NO_3_^−^ anions. After that, NO_3_-ISEs were re-filled with mixed solution containing 0.01 M KNO_3_ and 0.01 M KCl. The chloride salt was needed for reliable work of the internal Ag/AgCl electrode.

Solid-contact Cd^2+^-ISEs were prepared as follows: The working surface of glassy carbon rods was polished with sandpaper, then with diamond slurry (1 µm), and finally with 0.3 µm alumina. After that, the electrodes were placed for 5 min in 0.1 M HNO_3_, rinsed with DI water, with ethanol and again with DI water. The ion-to-electron transducer layer of polyethylendioxythiophene doped with polystyrenesulfonate (PEDOT-PSS) was deposited on GC galvanostatically from the mixed solution of EDOT and NaPSS as described elsewhere [24]. Next, membranes with thicknesses from 10 to 150 µm were deposited on the polymer layer by drop-casting 10–20 µL of cocktails with “dry” content of 2 to 10%.

Mixed solutions were prepared by adding suitable aliquots of stock (1 or 0.1 M) solutions of KCl, CaCl_2_, CdCl_2_, KNO_3_, and NaCl to a 500 mL volumetric flask, and filling the flask with deionized water. Solutions with low concentrations of KCl, CaCl_2_ and CdCl_2_ also contained NaCl to ensure reliable work of Ag/AgCl electrodes. The compositions of the solutions are presented in Table 2.

Zero current potentiometric measurements were performed with Ecotest-120 8-channel potentiometric station Econics (Moscow, Russia). The reference electrode was a single junction Ag/AgCl electrode in 3.5 M KCl, with a salt bridge with a limited leak of KCl. Calibrations were carried out from 0.1 M down to 10^−8^ M respective solution: KCl, CaCl_2_, CdCl_2_ or KNO_3_, using automatic burette Metrohm 700 Dosino controlled by Metrohm 711 Liquino Controller (Metrohm, Buchs, Switzerland).

Chronopotentiometric curves and electrochemical impedance spectra were recorded with Potentiostat-Galvanostat Autolab 302N with a frequency response analyzer module FRA 2 (Metrohm, Utrecht, The Netherlands).

The resistance of the membranes was studied by chronopotentiometric and impedance measurements. In chronopotentiometric measurements, two protocols were utilized: “slow” and “fast”. In the “slow” protocol, the open circuit potentials (OCPs) were recorded for the first 10 s, then the current value was abruptly changed from zero to 10^−7^ A (the respective current density was 1.27∙10^−7^ A/cm^2^), and the potential was registered for 60 s. After that, the current was turned off, and the potential was registered for another 60 s. The time resolution in the “slow” protocol was 0.2 s. In the “fast” protocol, the OCPs were recorded for the first 3 s, then the current value was abruptly changed from zero to 10^−7^ A and the potential was registered for 10 s, then the current was turned off, and the potential was registered for another 10 s. The time resolution in the “fast” protocol was 0.005 s.

The impedance measurements were made in potentiostatic mode with the excitation magnitude of ± 5 mV around the OCP, within a frequency range from 100 kHz to 0.1 Hz.

All measurements were carried out at room temperature (22 ± 1 °C). Three replicate samples with each membrane composition were used in this work.

Water uptake by the membranes was studied gravimetrically, in the same way as before [17,21]. Large pieces of membranes (ca 1 g) were placed into 50 mL beakers with solution for 1 week to be sure that the equilibrium is established (the equilibration time was actually 2–3 days). After 1 week in solution, the membranes were withdrawn from the beakers and the membrane surfaces were gently dried with tissue paper; they were then weighted giving mass $M_{1}$: that of the membrane equilibrated with solution. After that, the membranes were left to dry in air until constant mass $M_{2}$. Dried membranes were fixed on special hangers, so the membranes were open to air from both sides. Water fraction in the membranes was calculated as $M_{W}=\left(M_{1}-M_{2}\right)/M_{2}$. The procedure was repeated 3 times with each membrane and each solution to access the experimental error. Conditioning of the membranes in solutions, drying in air, and weighting were performed at room temperature (22 ± 1 °C).

The thickness of thin membranes deposited on solid-contact ISEs was measured with Olympus BX-51 (Tokyo, Japan) microscope. Membranes were detached from the ISEs, applied to a pyroceramic substrate, and bisected. Afterwards, photos were taken of the cross-section of the substrate with the membrane and processed with the microscope software; an example is shown in Appendix A.

## 3. Results

### 3.1. Potentiometric Response of the ISEs

For the study of the potentiometric response, the ISEs were filled, respectively, with 0.01 M KCl, 0.01 M CaCl_2_, 0.01 M CdCl_2_ and 0.01 MKNO_3_ + 0.01 M KCl. The reference electrode was a single junction Ag/AgCl electrode in 3.5 M KCl, with a salt bridge with a limited leak of KCl. The potentiometric measurements (see Figure 1) delivered the data on the slope, span, and selectivity of the ISEs.

One can see that the ISEs showed a linear response within the concentration range of the respective electrolytes from 0.1 M to 0.01 mM with near-Nernstian slopes. The EMF values recorded in 0.1 M NaCl confirmed the high selectivity of K-, Ca- and Cd-ISEs to their primary ions over Na^+^, and that of NO_3_-ISE to NO_3_^−^ over Cl^−^. These results confirmed the suitability of the electrodes for further studies.

### 3.2. Resistivity of the Membranes

Chronopotentiometric and impedance measurements with classical ISEs containing internal aqueous solution and internal Ag/AgCl electrode were performed in symmetric cells: external and internal solutions were the same, so ISE membranes were equilibrated with the same solution from both sides. The compositions of the solutions are presented in Table 2. The reference electrode was Ag/AgCl (chlorinated silver wire), and glassy carbon rod served as counter electrode.

Chronopotentiometric measurement of the membrane bulk resistance assumes calculation of the resistance as the ratio of the instant Ohmic drop over the current. In reality, the curve is recorded with a certain acquisition time, so the results obtained by two protocols “slow” and “fast” may be different. The values obtained by the “slow” protocol may include the beginning of the polarization or relaxation and therefore overestimate the resistance compared to the data obtained by the impedance measurements [15]. Examples of the chronopotentiometric curves obtained in “slow” and “fast” protocols are presented in Figure 2. The complete data on the consistency between the results obtained by “fast” and “slow” protocols are presented in Appendix A.

One can see that the polarization accumulated in the “slow” protocol is larger than that in the “fast” protocol, while the relaxation in the “slow” protocol is more complete than that in the “fast” protocol. However, Ohmic drops (when current is turned on or off) in the polarization curves were recorded in both protocols: “fast” and “slow” are practically the same, although the drop values measured with the “slow” protocol are ca. 2–3 % higher. The values of the membrane bulk resistance calculated as ratios of the Ohmic drops to the current value obtained using the two protocols are also the same within a 2–3 percent error respectively.

Examples of the impedance data (Nyquist plots) are presented in Figure 3.

High-frequency semicircles in the impedance spectra were fitted with a circuit containing a resistor R and a constant phase element CPE connected in parallel, and Warburg impedance W in series. The fitting was performed with the Autolab built-in software FRA 4.9.007; the χ^2^ values were ca. 10^−3^–10^−2^. The CPE values were ca. 5.0∙10^‒11^ F∙s^n−1^ for K- and NO_3_-ISEs, ca. 1.2∙10^‒10^ F∙s^n−1^ for Ca-ISEs, and ca. 1.4∙10^‒10^ F∙s^n−1^ for Cd-ISEs. Factor n varied from 0.88 to 0.95; thus, the CPEs were close to ideal capacitors. The obtained CPE values are consistent with those expected for the geometric capacitances of PVC membranes with non-polar plasticizers: BBPA (K-ISEs) and DOP (NO_3_-ISEs), and with polar oNPOE (Ca- and Cd-ISEs). The polarity of PVC membranes plasticized with oNPOE (ε ≈ 14) [25] is lower than that of pure oNPOE (ε = 24), but it is still higher than the polarity of membranes with BBPA or DOP. This allows attributing the high-frequency semicircles to the bulk of the membranes.

The data on the membrane bulk resistance obtained from the impedance spectra are consistent with those obtained from the chronopotentiometric curves (see Appendix A).

The data on the resistivity of the membranes equilibrated with mixed solutions are presented in Figure 4 (vs. the total concentration of ions, TCI) and in Figure 5 (vs. the ionic strength, IS). No regular change of the resistance can be seen over the range of TCI or IS from 0.1 to 0.001 M. However, with further dilution, the resistivity increases significantly. Thus, the curves are similar to those obtained earlier in pure electrolyte solutions containing the respective primary ions: CaCl_2_ [14], KNO_3_ [15], KCl [16] and CdCl_2_ [17]. In addition, the dependencies of the resistivity of the two factors TCI and IS are similar, although they shifted slightly relative to one another because TCI and IS, per se, do not coincide.

For measurements with dry membranes, disks with a diameter of 12 mm cut from master membranes were equilibrated for 1 week with 0.01 M solutions of electrolytes containing the respective primary ion and dried in air for 1 day. After that, discs were placed between 2 larger discs made of polished copper. One copper disc was connected to the instrument as the working electrode, the other as reference, and counter electrodes short-circuited.

Impedance measurements with dry membranes revealed capacitive behavior at low frequencies, consistent with large polarization in chronopotentiometric experiments (see Appendix A). This result was expected because the interface between ionically conducting membranes and metal copper is blocked and acts as a capacitor.

Importantly, high-frequency resistances in impedance measurements and Ohmic drops in chronopotentiometric measurements with dry membranes clearly show that the resistivity of dry membranes is significantly lower than that measured in contact with solutions (see Appendix A). This observation confirms the role of water uptake in the resistance of the ISE membranes.

### 3.3. Water Uptake

The results of the measurements of water uptake are presented in Figure 6 and in Appendix A. It can be seen that water uptake by membranes plasticized with oNPOE (Ca- and Cd-ISEs) is much larger than water uptake by membranes with BBPA (K-ISE) and DOP (NO_3_-ISE). This is most probably due to the difference in the polarity of the membranes.

Data shown in Figure 6, although noisy, show that water uptake by the membranes increases along with the decrease of the total concentration of ions. This result is consistent with the suggestion that water uptake is governed by the osmotic pressure of the solution.

### 3.4. Measurements with Thin Membranes

It was shown that the change of the membrane bulk resistance upon a change of the solution concentration is surprisingly fast. Most of the change in resistance happens within a few minutes after the replacement of the previous solution with the next one, and these changes are reversible [14,15,16,17]. This implies a special role of the membrane layer in the vicinity of the membrane/solution interface. It was shown, although indirectly, that with the use of colored additives, the outer layers of ISE membranes are enriched in water [26,27,28]. Recently, ATR-FTIR imaging of the distribution of PVC, plasticizer, and water in the vicinity of the membrane/solution interface directly showed that water is located primarily in the surface layers of the membrane, and at a depth ca. 150 microns, no water was registered [21].

On the other hand, values of the membrane bulk resistance refer to the whole membrane volume. Therefore, the data shown in Figure 4 and Figure 5 represent resistivities averaged over the whole volume of the membrane. The non-uniform distribution of water in ISE membranes suggests that effects caused by water uptake must be more pronounced for thin membranes. We therefore studied the resistances of membranes with thicknesses from 10 to 150 µm. Since Cd-ISE shows the largest effects in resistivity, this ISE was chosen as a model system for this part of the study. Thin membranes are not mechanically robust enough and require support. Therefore, for studies with thin membranes, we used ISEs in solid-contact construct, i.e., without internal aqueous solution.

The results are shown in Figure 7. One can see that the decrease of the membrane thickness results in increased sensitivity of the membrane resistance to the concentration of the electrolyte in solution. This observation is consistent with the assumption that the changes of the ISE membrane resistance are driven by water uptake.

## 4. Discussion of the Role of Water Droplets in the ISE Membrane Resistance

It is known that ISE membranes consisting of plasticized PVC sorb water from solutions, and water in these membranes forms droplets with a minimum diameter ca. 16 nm [22,29]. The existence of different states of water in membranes was detected by attenuated total reflection Fourier transform infrared spectroscopy (ATR-FTIR) [30], and by hyphenated ATR-FTIR spectroscopy and impedance studies [31,32].

Formation of water droplets means that ISE membranes are essentially heterogeneous, while according to the theory of ISEs, membranes are assumed to be homogeneous phases. By contrast, membranes’ structural features and non-homogeneity are considered crucial for the membrane performance for separation. Tuning of the architecture of hydrogel membranes was used to improve engineered osmosis [33]. Boron nitride nanosheets in polyvinylidene fluoride-co-hexafluoropropene membranes were incorporated for improved desalination performance [34], and aliphatic polyketone membranes were doped with alginate additive to accelerate or inhibit solvent-nonsolvent exchange rate and improve the hydrophilicity of the membranes [35]. It was also shown that the structural (surface heterogeneity) and material characteristics (yield stress) influence hydraulic resistance during filtration through polyethersulfone membranes [36].

Bearing in mind the presence of water droplets in ISE membranes, we tentatively explain the non-constancy of the membrane bulk resistance within the Nernstian response range as follows: Unlike the assumption in current theoretical descriptions, membranes must be considered heterogeneous two-phase materials containing a continuous organic phase and a dispersed aqueous phase. The electrochemical equilibrium is established between the solution and the continuous organic phase of the membrane, and therefore, as long as the composition of the organic phase is constant, the ISE shows linear Nernstian response to the activity of the analyte ion. However, the resistance is sensitive to the presence of the dispersed aqueous phase (water droplets). Complexes of ions with ionophores, as well as ion-exchanger ions, are lipophilic and are therefore confined to the organic phase of the membrane. Therefore, for a diffusing lipophilic ion, a water droplet forms an obstacle which must be passed around as shown schematically in Figure 8. Therefore, the mean length of the diffusion path of a lipophilic ion within a membrane increases along with an increase of water uptake. This is equivalent to the decrease of the mobilities of ions within the membrane. Because of this, the bulk resistance of the membrane is not constant and follows water uptake, although the ISE shows linear Nernstian response to the activity of the analyte ion.

## 5. Conclusions

This study confirms the data on the non-constancy of the bulk resistance of ISE membranes within the Nernstian response range obtained earlier [14,15,16,17]. For the first time, it is shown that the bulk resistance of ISE membranes, as well as water uptake, are determined by the ionic strength of the aqueous solution (or by the total concentration of electrolytes), rather than by the concentration of the respective primary ion. The proposed tentative explanation of the non-constancy of the membrane bulk resistance within the range of the Nernstian response of ISEs is based on the consideration of membranes as heterogeneous objects. This is in contrast with current theories of ISEs which suggest that membranes are homogeneous objects. However, accounting for heterogeneity does not call into question the mass of knowledge of ion–ionophore complexation and other processes in the organic phase proper, in particular the interpretation of the selectivity and the working range of the ionophore-based ISEs. Rather, accounting for the heterogeneity of the membranes allows for a more complete understanding of the membrane properties, in particular of the membrane resistance.

In principle, the dependence of the membrane resistance on the ionic strength of the solution opens a practically new relevant opportunity. In the potentiometric mode, ISEs deliver data on the activities of analytes. Obtaining data on the ionic strength via measurements of the resistance of ISEs allows calculation of the analyte concentration. Thus, the same sensor can be used for measurements of both activity and concentration of the analyte ion in solution. Obviously, this option requires further studies.

## Figures and Tables

**Figure 1 membranes-11-00344-f001:**
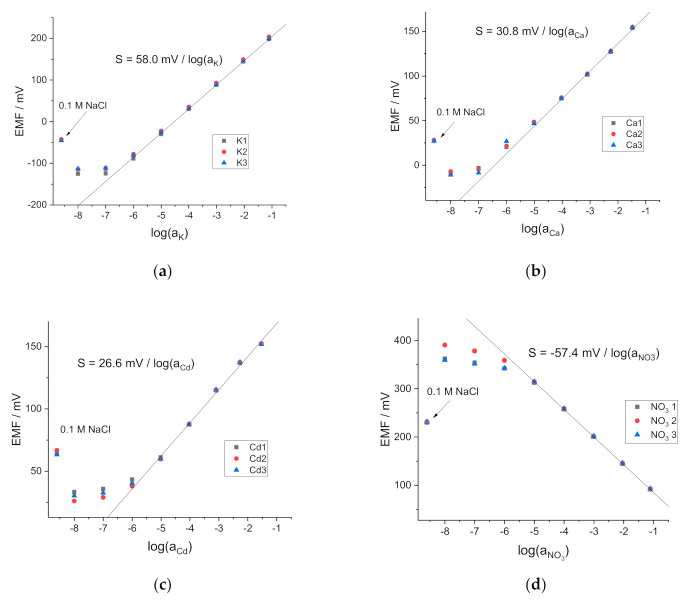
Calibration plots of the Ion-selective electrodes (ISEs). (**a**) K-ISEs 1, 2, 3 in KCl; (**b**) Ca-ISEs 1, 2, 3 in CaCl_2_; (**c**) Cd-ISEs 1, 2, 3 in CdCl_2_; (**d**) NO_3_-ISEs 1, 2, 3 in KNO_3_.

**Figure 2 membranes-11-00344-f002:**
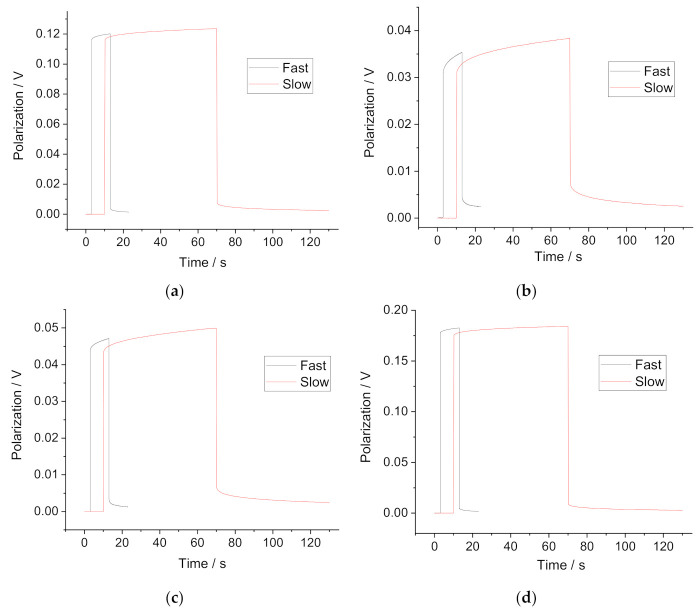
Examples of the chronopotentiometric curves obtained with “slow” and “fast” protocols in solution #10. (**a**) K-ISE; (**b**) Ca-ISE; (**c**) Cd-ISEs; (**d**) NO_3_-ISE.

**Figure 3 membranes-11-00344-f003:**
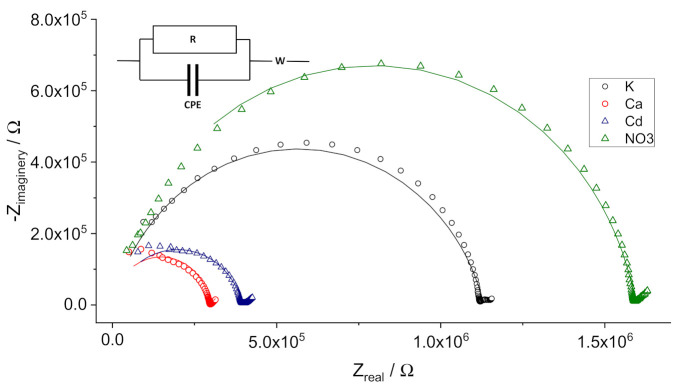
Nyquist plots of the impedance spectra of K-, Ca-, Cd- and NO_3_-ISEs in solution #1. Symbols: experimental data; solid lines: fitted data.

**Figure 4 membranes-11-00344-f004:**
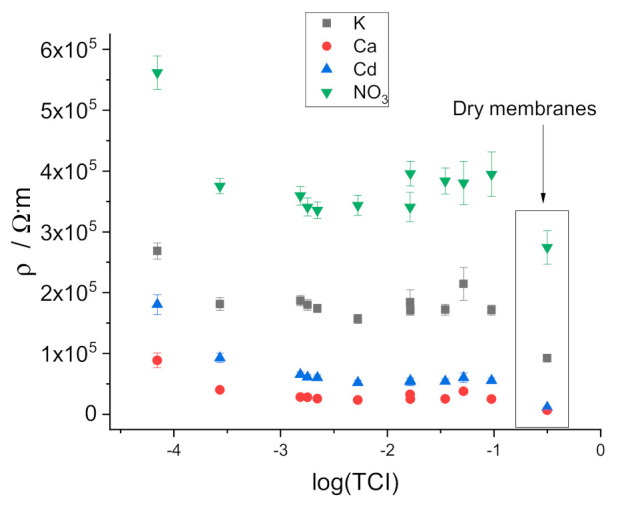
Dependence of the resistivity of the membranes on the total concentration of ions in mixed solutions.

**Figure 5 membranes-11-00344-f005:**
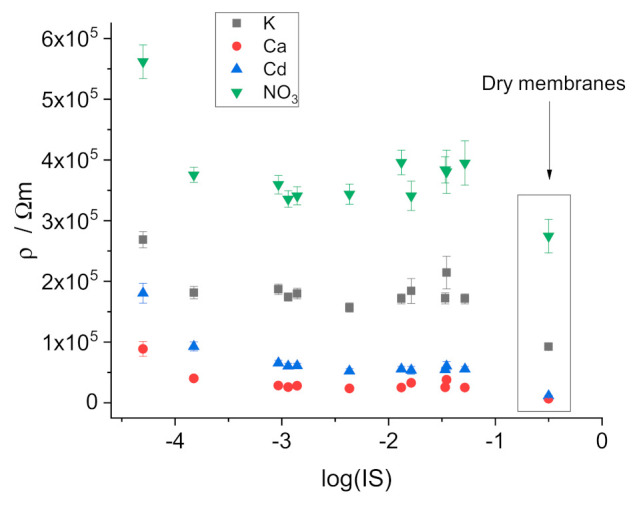
Dependence of the resistivity of the membranes on the ionic strength of mixed solutions.

**Figure 6 membranes-11-00344-f006:**
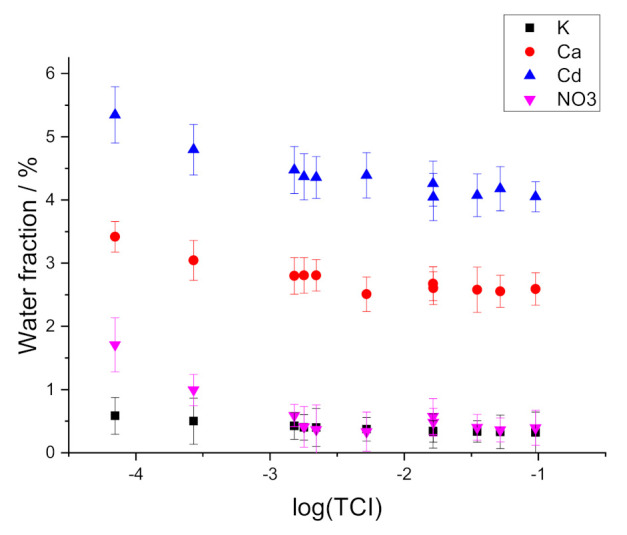
Dependence of water fraction in membranes on the total concentration of ions in solution.

**Figure 7 membranes-11-00344-f007:**
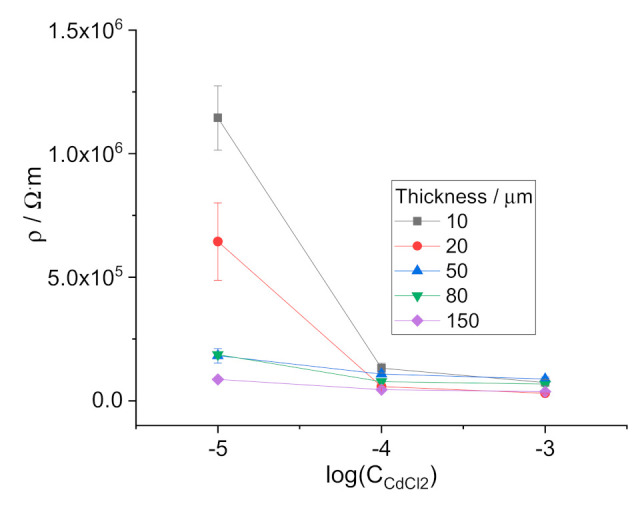
Resistivity of Cd^2+^-selective membranes vs. concentration of CdCl_2_.

**Figure 8 membranes-11-00344-f008:**
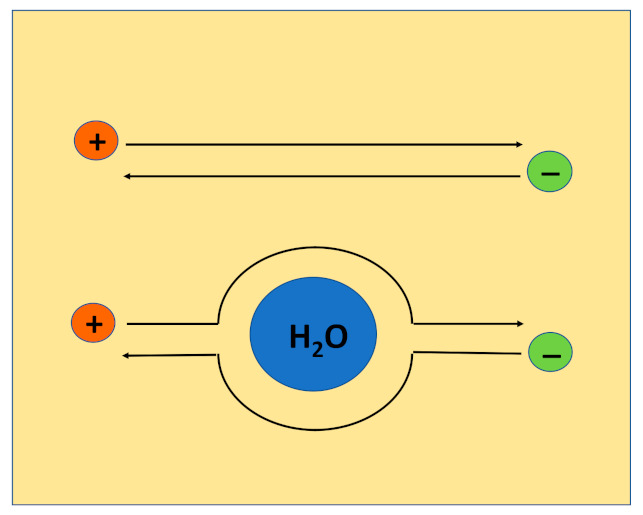
Ion paths without passing around obstacles (top) and with passing around water droplet (bottom).

**Table 1 membranes-11-00344-t001:** Membrane compositions (weight percentage and molality of the components).

Ion	Neutral Ionophore	Ion Exchanger	Plasticizer	PVC
K^+^	Valinomycin 1.5%, 0.02 M	KClTPB, 0.3%, 0.01 M	BBPA 65.5%	32.7%
Ca^2+^	ETH 1001 0.9%, 0.02 M	KClTPB 0.3%, 0.01 M	oNPOE 65.9%	32.9%
Cd^2+^	ETH 1062 0.6%, 0.02 M	KClTPB 0.3%, 0.01 M	oNPOE 66.1%	33.0%
NO_3_^−^	-	TDABr 2.2%, 0.05 M	DOP 65.2%,	32.6 %

**Table 2 membranes-11-00344-t002:** Compositions of the solutions: concentrations of salts, ionic strength (IS) and total concentration of ions (TCI).

Solution	Concentration, mM					log(IS)	log(TCI)
	KCl	CaCl_2_	CdCl_2_	KNO_3_	NaCl		
1	10.0	3.00	0.00	3.00	30.0	−1.28	−1.02
2	6.00	2.00	4.00	1.00	10.0	−1.46	−1.28
3	0.00	1.00	10.0	1.00	0.00	−1.47	−1.46
4	3.00	0.30	3.00	0.30	0.00	−1.88	−1.78
5	0.10	0.40	5.00	0.00	0.00	−1.79	−1.79
6	1.00	0.10	1.00	0.00	0.00	−2.37	−2.28
7	0.03	0.00	0.03	0.03	1.00	−2.94	−2.66
8	0.30	0.03	0.30	0.10	0.00	−2.86	−2.75
9	0.10	0.01	0.10	0.00	0.50	−3.03	−2.82
10	0.01	0.00	0.01	0.01	0.10	−3.82	−3.57
11	0.01	0.01	0.01	0.01	0.00	−4.30	−4.15

## Data Availability

The data presented in this study are available in this article, and corresponding Appendix A.

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
