# Peer review of "The Origin of the Non-Constancy of the Bulk Resistance of Ion-Selective Electrode Membranes within the Nernstian Response Range"

_membranes, 2021, doi:10.3390/membranes11050344_

Round 1

Reviewer 1 Report

In the manuscript, the authors describe the bulk resistance of ion-selective membranes. They used four different types of ISE (the electrodes selective to K+, Ca2+, Cd2+ and NO3-, respectively), and established the resistance using two electrochemical methods, namely chronopotentiometry and impedance spectroscopy. Authors discussed the relation between the bulk resistance of the membrane and the parameter of the solution (ionic strength as well as the total ion concentration). Finally, they gravimetrically measured the water uptake and discussed it's influence on the bulk resistance.

The obtained results suggest, that in certain cases the membrane cannot be treated as homogeneous object, hence showing the limitations of existing models.

The manuscript has a very high scientific value, however there are several issues that should be addressed:

- It should be clarified if the concentrations mentioned in the caption of table 2 (and the text of the manuscript) refer to inner solution or the analyte solution.

- in lines 201-255, it is stated that values obtained using two chronopotentiometric protocols are the same within 2-3% error. These values are not shown. They should be included either in the manuscript or in the supporting information.

- in line 213 authors mention that spectra were fitted with equivalent circuit. The method used for curve fitting and the equation for the goal function should be given. The values of the goal functions for the fitted EIS in Fig. 3 (and S3), should be mentioned as well.

- Does the Table S3 contain the chronoamperometric values obtained using "slow" or "fast" protocol? Please clarify.

- The rows in Tables S1 and S2 are in the order of increasing total concentration, while the order in Table 2 seems random. Please unify this, so it is easier for the reader to make a connection between the particular concentration sets and the results.

Author Response

Replies to Reviewer 1

We highly acknowledge interesting and useful comments raised by the Reviewer, and we tried our best to improve the manuscript accordingly.

To make the modifications visible we used the TRACK CHANGE option.

Below, we answer the Reviewer’ comments point by point.

Comment

- It should be clarified if the concentrations mentioned in the caption of table 2 (and the text of the manuscript) refer to inner solution or the analyte solution.

Answer

Actually, it was stated in the Materials and Methods section that the chronopotentiometric and impedance measurements were performed with symmetric cells i.e. the internal and external solutions were the same. However, we agree that it is better to state this clearly in the Results section. This is done (lines 195-200) in the revised manuscript), and the respective part in the Materials and Methods section (lines 150-160 in the revised manuscript) is deleted to avoid repetition.

Comment

- in lines 201-255, it is stated that values obtained using two chronopotentiometric protocols are the same within 2-3% error. These values are not shown. They should be included either in the manuscript or in the supporting information.

Answer

We added the respective table to Supplementary Materials: Table S1.

Comment

- in line 213 authors mention that spectra were fitted with equivalent circuit. The method used for curve fitting and the equation for the goal function should be given. The values of the goal functions for the fitted EIS in Fig. 3 (and S3), should be mentioned as well.

Answer

The fitting was performed with the Autolab built-in software FRA 4.9.007, the χ2 values were ca. 10−3 - 10−2. This is now explained, lines 227, 228 in the revised manuscript.

Comment

- Does the Table S3 contain the chronoamperometric values obtained using "slow" or "fast" protocol? Please clarify.

Answer

It is now stated clearly in the Table heading that the values refer to “fast” protocol.

Comment

- The rows in Tables S1 and S2 are in the order of increasing total concentration, while the order in Table 2 seems random. Please unify this, so it is easier for the reader to make a connection between the particular concentration sets and the results.

Answer

The order in Table 2 (original manuscript) referred to the actual order of the measurements. The order was randomized deliberately: to minimize possible memory effects. In the revised manuscript, the order of the rows in all tables is unified, and refers to the total concentration of ions in solutions, from high to low.

Reviewer 2 Report

This is quite simple paper which studies ion selective electrodes by means of few electrochemical methods. Authors found out that when ISE membrane is in contact with diluted solutions it absorbs water. The phenomenon is known for many years. It is also found that due to water absorption the ISE membrane resistances increase. Then authors formulated the hypothesis that it is due to water droplets which impedes the movement of hydrophobic ions in the membrane. However, the weak point of ISE solid contact electrodes is peeling off the ISE membrane from the conductive solid electrode due to water diffusion across the ISE membrane, which is responsible for short life of these electrodes in watery medium. Could author prove that this was not the case in their experiment? I would recommend rewriting the paper. Language improvement is recommended. The word “coherent” should not be used in the paper’s context.

Author Response

Replies to Reviewer 2

We highly acknowledge interesting and useful comments raised by the Reviewer, and we tried our best to improve the manuscript accordingly.

To make the modifications visible we used the TRACK CHANGE option.

Below, we answer the Reviewer’ comments point by point.

Comment

Authors found out that when ISE membrane is in contact with diluted solutions it absorbs water. The phenomenon is known for many years.

Answer

Yes, the phenomenon per se is known for long and we did our best referring the relevant papers published on this issue: refs. 22, 23, 26-32. However, the dependence of the water uptake on the concentration of pure electrolyte solutions was studied systematically for the first time in our earlier study (ref. 17), and the dependence of the water uptake on the total concentration of ions and on the ionic strength in mixed solutions is reported in this manuscript for the first time. This is stated in the Introduction. We believe that these dependencies are non-trivial since the activity of water in all these solutions approaches unity, and approaches closer along the dilution, whereas the dependence of the water uptake gets sharper along the dilution.

Comment

It is also found that due to water absorption the ISE membrane resistances increase. Then authors formulated the hypothesis that it is due to water droplets which impedes the movement of hydrophobic ions in the membrane. However, the weak point of ISE solid contact electrodes is peeling off the ISE membrane from the conductive solid electrode due to water diffusion across the ISE membrane, which is responsible for short life of these electrodes in watery medium. Could author prove that this was not the case in their experiment?

Answer

We agree that the membranes of solid-contact ISEs sometimes peel off from the substrate (or from the transducer layer). This did not happen in our case. Even more important is that most of the results reported here were obtained with classical ISEs containing internal aqueous solution and internal Ag/AgCl electrode.

Comment

I would recommend rewriting the paper. Language improvement is recommended.

Answer

We did our best clarifying several issues, and made several corrections in the revised manuscript.

Comment

The word “coherent” should not be used in the paper’s context.

Word “coherent” is now replaced with word “continuous” throughout the manuscript.

Round 2

Reviewer 2 Report

The paper is slightly improved. It may be accepted in present form.